# One-Year Hemodynamic Performance of Three Cardiac Aortic Bioprostheses: A Randomized Comparative Clinical Trial

**DOI:** 10.3390/jcm10225340

**Published:** 2021-11-16

**Authors:** Lourdes Montero-Cruces, Manuel Carnero-Alcázar, Fernando José Reguillo-Lacruz, Francisco Javier Cobiella-Carnicer, Daniel Pérez-Camargo, Paula Campelos-Fernández, Luis Carlos Maroto-Castellanos

**Affiliations:** Department of Cardiac Surgery, Cardiovascular Institute, Hospital Clínico San Carlos, 28940 Madrid, Spain; mcarneroalcazar@gmail.com (M.C.-A.); fjrl1965@yahoo.es (F.J.R.-L.); jcobiella@gmail.com (F.J.C.-C.); daniel.perezc@gmail.com (D.P.-C.); paulacampelos@hotmail.com (P.C.-F.); lcmarotoc@hotmail.com (L.C.M.-C.)

**Keywords:** cardiac surgery, aortic valve replacement, bioprostheses, heart valve

## Abstract

Background: We aimed to compare 1 year the hemodynamic in-vivo performance of three biological aortic prostheses (Carpentier Perimount Magna Ease^TM^, Crown PRT^TM^, and Trifecta^TM^). Methods: The sample used in this study comes from the “BEST-VALVE” clinical trial, which is a phase IV single-blinded randomized clinical trial with the three above-mentioned prostheses. Results: 154 patients were included. Carpentier Perimount Magna Ease^TM^ (*n* = 48, 31.2%), Crown PRT^TM^ (*n* = 51, 32.1%) and Trifecta^TM^ (*n* = 55, 35.7%). One year after the surgery, the mean aortic gradient and the peak aortic velocity was 17.5 (IQR 11.3–26) and 227.1 (IQR 202.0–268.8) for Carpentier Perimount Magna Ease^TM^, 21.4 (IQR 14.5–26.7) and 237.8 (IQR 195.9–261.9) for Crown PRT^TM^, and 13 (IQR 9.6–17.8) and 209.7 (IQR 176.5–241.4) for Trifecta^TM^, respectively. Pairwise comparisons demonstrated improved mean gradients and maximum velocity of Trifecta^TM^ as compared to Crown PRT^TM^. Among patients with nominal prosthesis sizes ≤ 21, the mean and peak aortic gradient was higher for Crown PRT^TM^ compared with Trifecta^TM^, and in patients with an aortic annulus measured with metric Hegar dilators less than or equal to 22 mm. Conclusions: One year after surgery, the three prostheses presented a different hemodynamic performance, being Trifecta^TM^ superior to Crown PRT^TM^.

## 1. Introduction

The Carpentier Perimount Magna Ease^TM^, Crown PRT^TM^, and Trifecta^TM^ (Figure 1) bovine pericardial valves are widely used worldwide. Nevertheless, there are few studies properly designed to compare the hemodynamic performance of these three bioprostheses.

There are different factors that make this comparison difficult to achieve: the heterogeneity of the sizers of the different prostheses, the disproportion between the measurements offered by the sizers of the different manufacturers, the real dimensions of the prostheses, the variability of echo assessment, and the variety in the anti-calcification treatment and design of the bioprostheses. The only way to obtain reliable information on the performance of the different bioprostheses, considering the factors described above, is by analyzing the hemodynamic performance in vivo [1].

Therefore, we designed a randomized clinical trial that aimed to investigate the hemodynamic and clinical outcomes of patients receiving any of the three aforementioned bioprostheses.

## 2. Materials and Methods

### 2.1. Study Design

The sample used in this study comes from the clinical trial “BEST-VALVE” (Comparison of 3 contemporary cardiac bioprosthesis: mid-term valve hemodynamic performance) registered in the EudraTC (European Database of Clinical Trials) with registration number: 2018-001658-87. Briefly, the “BEST-VALVE” clinical trial was a single-center randomized phase IV clinical trial, with observer-blind analysis, prospective, and longitudinal, with 3 bovine pericardial aortic bioprostheses: Carpentier Perimount Magna Ease™ (Edwards LifeSciences Corporation©, Irvine, CA, USA), Crown PRT^TM^ (LivaNova, Saluggia, Italy), and Trifecta™ (Abbott, IL, USA). This study was approved by the local Ethics Committee. Informed consent for the study was obtained from every patient before randomization.

The study included patients undergoing aortic valve replacement at our hospital from June 2014 to June 2017 with random assignment (1:1:1) of the type of bioprosthesis. Initially, a sample size of 396 patients was estimated to achieve a power estimation of 80%, assuming a 10% loss. Due to the commercialization of the new Trifecta with glide technology and a new anti-calcification treatment (AC) Linx™ in 2016, the commercialization of the initial model of the prosthesis was discontinued. Consequently, the study was interrupted in 2017, when the prosthesis was no longer available at our center. Finally, a total of 154 patients were included in the study.

The different types of variables included in the study can be classified into pre, intra, and postoperative, and follow-up variables. Echocardiographic variables were collected before and after the procedure (echocardiographic information was recorded in the immediate postoperative period prior to discharge, 1month, 6 months, and 12 months after the procedure). The follow-up was performed throughout the outpatient clinic with the same interval times.

The primary endpoint was to compare the hemodynamic performance (aortic mean and peak gradients, peak aortic velocity, and effective orifice area) quantified by in-vivo echocardiogram 12 months after the implantation of 3 biological aortic prostheses (Carpentier Perimount Magna Ease^TM^, Crown PRT^TM^, and Trifecta^TM^).

Secondary endpoints included a stratified comparison of the primary endpoint according to the bioprostheses size and by the diameter of the aortic annulus (measured with a metric Hegar dilator); and a comparison between the 3 types of bioprostheses, including survival and the survival from a composite event (death, stroke, myocardial infarction, thromboembolism, endocarditis, and aortic valve reintervention) at 12-month follow-up.

### 2.2. Patient Inclusion and Exclusion Criteria

Inclusion criteria were patients over 18 years of age with the diagnosis of aortic stenosis or regurgitation and with an indication for valve replacement according to the recommendations of the European Society of Cardiology Guidelines on Valvular Heart Disease or American College of Cardiology Guidelines [2,3].

Exclusion criteria were pregnancy, concomitant pathology of the ascending aorta or acute aortic syndrome requiring a surgical repair, concomitant surgery of the left ventricular outflow tract or any other valve, aortic annulectasia (diameter of the aortic annulus measured with metric Hegar dilators > 25 mm), small aortic annulus (diameter of the aortic annulus measured with a metric Hegar dilators < 19 mm), active endocarditis, urgent or emergent surgery, redo procedures, an impossibility for physical, mental, or social reasons to achieve the study follow-up protocol or participation in any other clinical study at the time of inclusion.

### 2.3. Surgical Aortic Valve Replacement

The interventions were performed by all surgeons who were members of the cardiac surgery team at our hospital, who followed the same surgical technique.

All interventions were performed through a full or mini mid-sternotomy, depending on the surgeon’s preferences. The aortic valve leaflets were excised, and the aortic annulus was decalcified. The native aortic annulus was measured with a metric Hegar dilator. The patient was randomized once the operator verified intraoperatively that anatomic exclusion criteria were not present. After randomization, the type of prosthesis to be implanted was communicated to the surgeon, and the size of the assigned prosthesis was selected using aortic sizers supplied by the valve manufacturers. The prosthesis was implanted with non-everting pledget stitches by a standard supra-annular position.

### 2.4. Follow-Up

A total of 6 visits were programmed for all the patients included in the study (preoperative, surgery, discharge, 1 month, 6 months, and 1 year after surgery).

### 2.5. Echocardiography

Echocardiographic evaluations were performed by certified echocardiographers, and the images were archived in digital format. Two-dimensional ecocardiography was employed for anatomical and morphological evaluation of the aortic valve in order to define the etiology of the aortic valve disease. Continuous Doppler measurements were employed to obtain the peak aortic velocity. Afterward, the peak aortic gradient and the mean aortic gradient were estimated by applying the Bernoulli formula. The continuity equation (Gorlin’s formula) was used to estimate the effective orifice area. End-diastolic and end-systolic volumes, as well as left ventricular ejection fraction, were calculated using the Simpson biplane method in an apical view. The left ventricular mass was determined according to the method of the American Association of Echocardiography modified by Devereux.

### 2.6. Statistical Analysis

The Shapiro–Wilk normality test was applied to assess the normality for continuous variables. Continuous variables were expressed as mean and standard deviation or median and interquartile range (IQR) as appropriate. Categorical variables were expressed as the absolute and relative frequency (%). The comparison of quantitative variables was assessed using an analysis of variance (ANOVA) or the Kruskal–Wallis test for independent samples. A multiple comparisons test was performed applying the Bonferroni correction. Categorical variables were compared using the chi-squared or Fisher’s exact test. Subgroups analyses were performed: (1) based on the prosthesis size, 2 study groups were formed (one including patients with prostheses less than or equal to 21 mm and the other one involving patients with prostheses greater than 21 mm). (2) Another comparison of patients was stratified by the size of the aortic annulus measured by a metric Hegar dilator (two study groups were formed, one with an aortic annulus less than or equal to 22 mm and the other one with an aortic annulus greater than 22 mm). Differences were considered statistically significant at *p*-values < 0.05. The incidence of postoperative adverse events was compared using multivariable logistic regression. Survival was estimated using the Kaplan–Meier method and comparison between groups with the log-rank test. An intention-to-treat analysis was performed, but none of the patients had a different prosthesis implanted than the one assigned at randomization, and there were no dropouts.

Statistical analysis was performed using Stata 15 (StataCorp 2015. College Station, TX, USA).

## 3. Results

A total of 154 patients who underwent aortic valve replacement in our center were included in the study. The Carpentier Perimount Magna Ease^TM^ prosthesis was implanted in 48 patients (31.2%), the Crown PRT^TM^ prosthesis in 51 patients (32.1%), and the Trifecta^TM^ prosthesis in 55 patients (35.7%). The median age of our study population was 76.5 years (IQR 71.5–79.5), and 92 patients (59.7%) were male. The median of EuroSCORE I and EuroSCORE II was 6.2% (IQR 4.4–8.5) and 2.3% (IQR 1.4–4), respectively. Most of the patients underwent elective surgery (70.8%).

Table 1 shows baseline characteristics in the different study groups. There were no statistically significant differences between groups, except for the priority of surgery.

According to the echocardiographic characteristics in the preoperative period, 135 patients (87.7%) had severe aortic stenosis, and only 15 patients (9.7%) had severe aortic regurgitation. The left ventricular ejection fraction was preserved in most of the patients 90.26% (*n* = 139). Table 1 compares the main echocardiographic variables analyzed in the preoperative period.

### 3.1. Intraoperative Data

The surgery was performed through mini-sternotomy in 23 patients (14.9%), and concomitant coronary artery bypass grafting was performed in 36 patients (23.4%). The most frequent prosthesis size was 21 mm (38.3%), followed by 23 mm (37.7%). The size of the aortic annulus measured by the metric Hegar dilator was 23 mm in 46 patients (29.9%). The remaining operative characteristics are described in Table 1. No statistically significant differences were found in operative variables between groups.

### 3.2. Perioperative Outcomes

The median stay in the intensive care unit was 1 day (IQR 1–3). Four patients (2.6%) died during admission. Of these, three patients had a Crown PRT^TM^ prosthesis implanted and one patient a Trifecta^TM^ prosthesis. No statistically significant differences were found in hospital mortality between groups (*p* = 0.211).

### 3.3. Echocardiographic Data 12 Months after Surgery

Although all three bioprostheses were associated with excellent hemodynamics performance at 12 months after the procedure, the peak aortic gradient, the mean aortic gradient, and the peak aortic velocity were significantly different between the different prostheses. Table 2 shows the main echocardiographic data twelve months after surgery.

Multiple comparisons analysis with Bonferroni’s correction showed that the peak aortic gradient was lower for Trifecta^TM^ prosthesis as compared with Crown PRT^TM^ prosthesis (*p* = 0.03), and the mean aortic gradient was lower for the Trifecta^TM^ prosthesis than for the Crown PRT^TM^ prosthesis (*p* = 0.03) (Table 2). There were no statistically significant differences between Trifecta^TM^ and Carpentier Perimount Magna Ease^TM^, and neither between Crown PRT^TM^ and Carpentier Perimount Magna Ease^TM^. Figure 2 depicts a Box Plot of the peak and mean gradient at 12 months after surgery stratified by type of prosthesis.

### 3.4. Subgroup Analyses

Based on the size of the implanted prosthesis, no statistically significant differences were found for the peak aortic gradient, mean aortic gradient, and peak velocity between the different prostheses implanted with a size greater than 21 mm. However, statistically significant differences were found between the different prostheses in patients who received a small prosthesis less than or equal to 21 mm (see Table 3).

Pairwise comparisons in patients with prostheses sizes ≤ 21mm, the peak aortic gradient for Trifecta^TM^ was significantly lower than for Crown PRT^TM^ (*p* = 0.004), and the mean aortic gradient was significantly lower for Trifecta^TM^ than for Crown PRT^TM^ (*p* = 0.014). The peak aortic velocity and the effective orifice area did not show statistically significant differences (Table 3). There were no statistically significant differences between Trifecta^TM^ and Carpentier Perimount Magna Ease^TM^, and neither between Crown PRT^TM^ and Carpentier Perimount Magna Ease^TM^.

Figure 3 depicts a Box Plot with the differences in the distribution of the peak and mean aortic gradients of the echocardiogram at the 12 months after surgery stratified by type and size of prosthesis.

Based on the size of the aortic annulus, no statistically significant differences were found for peak aortic gradient, mean aortic gradient, and peak velocity between implanted prostheses with an aortic annulus greater than 22 mm. However, statistically significant differences were found between the different prostheses in patients with an aortic annulus less than or equal to 22 mm (Table 4).

In patients with an aortic annulus size less than or equal to 22 mm, the peak aortic gradient for Trifecta^TM^ was significantly lower than Crown PRT^TM^ (*p* = 0.006), the mean aortic gradient was significantly lower for Trifecta^TM^ than Crown PRT^TM^ (*p* = 0.018), and the peak aortic velocity was also significantly lower for Trifecta^TM^ than Crown PRT^TM^ prosthesis (*p* = 0.048) (Table 4).

### 3.5. Postoperative Events at 12 Months

No statistically significant differences were found between the three prostheses in the incidence of postoperative events at 12 months after the procedure (Table 5). A multivariable logistic regression analysis of postoperative events at 12 months was performed between the three types of prosthesis adjusted for preoperative priority of surgery. No differences were observed between groups.

### 3.6. Survival at 12 Months

Survival rate was 98.7%, 94.8%, and 92.8% at 1 month, 6 months, and 12 months, respectively. One year survival rate was 97.9% for patients with a Carpentier Perimount Magna Ease^TM^ prosthesis, 94.4% for patients with a Trifecta^TM^ prosthesis, and 86.3% for patients with a Crown PRT^TM^ prosthesis (Figure 4A). No statistically significant differences were found between the different prostheses (Log rank *p* = 0.0647).

A subgroup survival rate comparison was made, observing statistically significant differences between patients with Carpentier Perimount Magna Ease^TM^ and Crown PRT^TM^ prostheses (94.4% vs. 86.3%, respectively, Log-rank *p* = 0.0338).

Survival from the composite event at 1 month, 6 months, and 12 months was 94.1%, 88.2%, and 85.7%, respectively. Event-free survival at 12 months was 89.6% for the Magna Ease^TM^ prosthesis, 82.4% for Crown PRT^TM^ prosthesis, and 85.5% for Trifecta^TM^ prosthesis. No statistically significant differences were found between the three groups (Log-rank *p* = 0.3564) (Figure 4B). A comparison of event-free survival by subgroups was also made, with no statistically significant differences between groups found.

## 4. Discussion

Aortic stenosis is the third most common cardiovascular disease after arterial hypertension and coronary disease [4,5]. Among the different types of valvular heart diseases, aortic stenosis is the most frequent, with a prevalence of 2% in people over 65 years of age and 10% in those over 80 years of age. Furthermore, a prevalence of aortic sclerosis from 26% to 34% was observed in people over 65 years of age [6].

Aortic valve replacement is the only treatment that can reduce symptoms, improve quality of life, and increase patient survival rate due to the possibility of reverse remodeling of the left ventricle and the regression of compensatory ventricular hypertrophy [7,8,9,10].

As a result of the progressive aging of the population and the preference of this subgroup of population for biological prostheses, due to their lower hemorrhagic and thromboembolic risk, a significant increase in the use of bioprostheses was observed in the last decades [11,12,13].

Although the echocardiogram is the technique of choice for the evaluation of valvular heart disease, magnetic resonance imaging has evolved to become a viable non-invasive alternative technique to echocardiography for a wide variety of cardiac conditions, including aortic prosthetic evaluation. It has the advantage that contrast is not required for basic functional assessment or for quantification of transvalvular flow, and it should be taken into account for future valvular heart studies [14,15]. Other techniques, such as computed tomography, are useful in evaluating the severity of valvular heart disease as well as pre-procedural planning, mainly prior to TAVI implantation [2,16].

In our study, three of the most widely implanted bioprostheses at the time of the study were selected. These bioprostheses were Trifecta^TM^, Carpentier Perimount Magna Ease^TM^, and Crown PRT^TM^.

The sample of patients in our study comes from a randomized clinical trial implemented in our center, being a representative sample of patients with aortic valve disease from the general population. A total of 154 patients who were undergoing aortic valve replacement were included. Due to the increase in the use of transcatheter aortic prostheses and its progressive extension to intermediate and low-risk patients, such a sample is a difficult achievement to accomplish in a single-center randomized clinical trial.

Adequate control of the variables was performed between the three groups. The population of patients who received Trifecta^TM^, Carpentier Perimount Magna Ease^TM^, and a Crown PRT^TM^ was similar, comparing the sociodemographic variables and the risk profile.

There are few studies comparing the hemodynamic performance of the different third-generation stent bioprostheses. The primary endpoint in our study was a comparison of the mean aortic gradient, the peak aortic gradient, the peak aortic velocity, and the effective orifice area quantified by echocardiography at 12 months after the surgery.

The TRIBECA study [17], a multicenter retrospective study that compared the postoperative hemodynamic results of the Trifecta^TM^ and Carpentier Perimount Magna Ease^TM^ prostheses, with a total of 791 patients analyzed, presented a median, mean aortic gradient of 10 mmHg (IQR: 8–13) for the Trifecta^TM^ prosthesis and 16 mmHg (IQR: 11–22) for the Carpentier Perimount Magna Ease^TM^ prosthesis at 12 months after the intervention. In addition, these statistically significant differences were maintained in all valve sizes.

Suri et al. [18], in their randomized clinical trial of 300 patients receiving a Carpentier Perimount Magna Ease^TM^, Mitroflow^TM^, or Epic^TM^ prosthesis implant, presented statistically significant better hemodynamic results in patients with a Carpentier Perimount Magna Ease^TM^ prosthesis with aortic rings greater than 23 mm in the immediate postoperative echocardiogram.

In our study, we observed statistically significant differences in the peak aortic gradient, the mean aortic gradient, and the peak aortic velocity between the three study groups in the echocardiogram performed at 12 months after the procedure. The echocardiogram confirmed that the hemodynamic performance with Trifecta^TM^ was noticeably better in comparison with Crown PRT^TM^, presenting statistically significant differences.

These differences were observed in small prostheses with a size equal to or less than 21 mm. In these cases, the mean and peak aortic gradients of Trifecta^TM^ were significantly lower than those of Crown PRT^TM^ (Peak and mean gradient of Trifecta^TM^ 12.3 mmHg (IQR 9.3–19.9) and 6.7 mmHg (IQR 4.3–10.5); vs. peak and mean gradient of Crown PRT^TM^ 26 mmHg (IQR 19.4–34.3) and 12 mmHg (IQR 9.4–18.0)).

Bach et al. [19], in their clinical trial comparing patients with Trifecta^TM^, Carpentier Perimount Magna Ease^TM^, and Freestyle^TM^ prostheses, described hemodynamically significant differences between the Carpentier Perimount Magna Ease^TM^ and Trifecta^TM^ prosthesis in favor of Trifecta^TM^ according to the echocardiogram performed at 10 months after the procedure.

The challenge of performing a comparison between the different types of prostheses lies in the variability of the prosthetic dimensions within the same size between the different manufacturers. In order to compare the true hemodynamic performance of the different bioprostheses, the study should be performed considering the diameter of the aortic annulus, for which a metric Hegar dilator can be used. One of our secondary endpoints was the stratified comparison of the mean gradient, the peak gradient, the peak velocity, and the effective orifice area, according to the diameter of the aortic annulus at 12 months after the intervention.

The differences observed between the study groups were also obtained after performing a comparison of the size of the aortic annulus. In this case, we obtained statistically significant differences regarding the peak gradient, the mean gradient, and the peak aortic velocity in Trifecta^TM^ compared to Crown PRT^TM^ for aortic annulus less than or equal to 22 mm.

In their clinical trial of 100 patients, Van Linden et al. [20] revealed a better hemodynamic performance in patients with Trifecta^TM^ compared to patients with Carpentier Perimount Magna Ease^TM^ in aortic annulus greater than or equal to 23 mm measured by the Hegar dilator in the echocardiogram performed in the immediate postoperative.

Fiegl et al. [21] also presented lower mean aortic gradients at one year after the procedure in their retrospective study in patients with Trifecta^TM^ compared to patients with Carpentier Perimount Magna Ease^TM^. However, these results were extrapolated to all aortic annulus sizes.

Survival rates for patients with Trifecta^TM^ prostheses in the group by Kilic et al. [22] was 94.9%, 89.7%, and 69.8% at 30 days, 1 year, and 5 years, respectively.

In our study, the highest survival rate was observed in patients with Carpentier Perimount Magna Ease^TM^. A subgroup survival analysis found statistically significant differences in survival rate between patients with Carpentier Perimount Magna Ease^TM^ and Crown PRT^TM^. In our series, the survival rate at 30 days and one year was 98.2% and 94.4% for Trifecta^TM^, 98.0% and 86.3% for Crown PRT^TM^, and 97.9% and 97.9% for Carpentier Perimount Magna Ease^TM^.

### Limitations

Despite being a randomized clinical trial, the main limitations in our study were associated with the fact that it was a single-center study with limited sample size. Therefore, the results obtained could not be comparable with those obtained in other centers and could interfere with the capacity to detect statistically significant differences due to the lack of sufficient statistical power.

## 5. Conclusions

There are significant differences in the hemodynamic performance between Carpentier Perimount Magna Ease^TM^, Crown PRT^TM^, and Trifecta^TM^ aortic bioprostheses. Mean and Peak aortic gradients 12 months after the intervention was significantly different between the three groups, being lower for Trifecta^TM^ compared to Crown PRT^TM^. These differences were found in small prostheses with a size less than or equal to 21 mm, as well as in patients with an aortic annulus less than or equal to 22 mm.

## Figures and Tables

**Figure 1 jcm-10-05340-f001:**
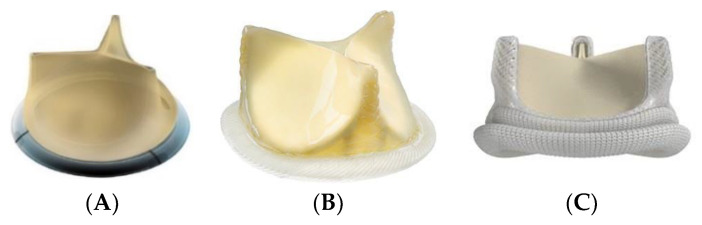
Pericardial aortic bioprostheses. Crown PRT^TM^. LivaNova (**A**); Trifecta^TM^. Abbott (**B**); Carpentier Perimount Magna Ease^TM^. Edwards Lifesciences (**C**).

**Figure 2 jcm-10-05340-f002:**
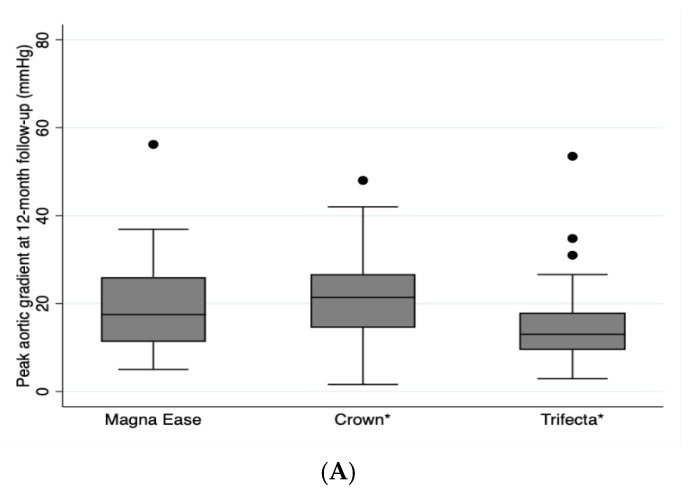
Depicts a Box Plot of the peak aortic gradient (**A**) and the mean aortic gradient (**B**) at 12 months after surgery stratified by type of prosthesis. * *p* = 0.003.

**Figure 3 jcm-10-05340-f003:**
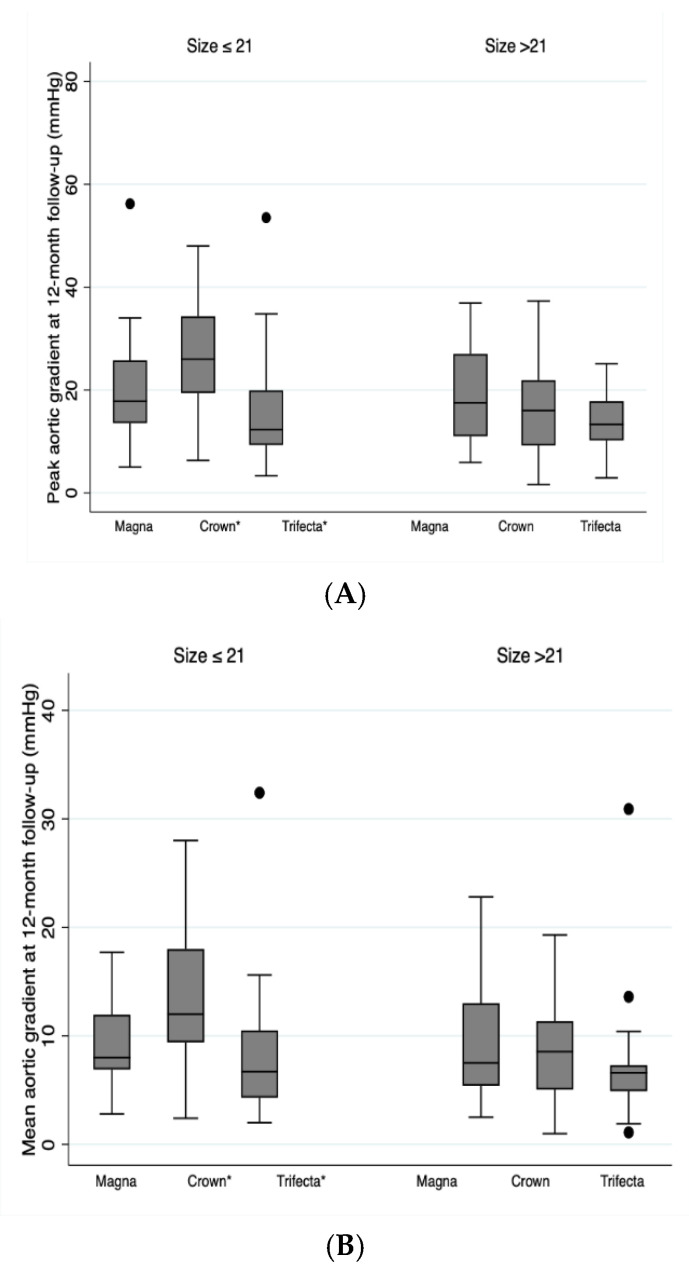
Box Plot for peak aortic gradient (**A**) and the mean aortic gradient (**B**) at 12 months after the surgery by size and type of prosthesis. * *p* = 0.001.

**Figure 4 jcm-10-05340-f004:**
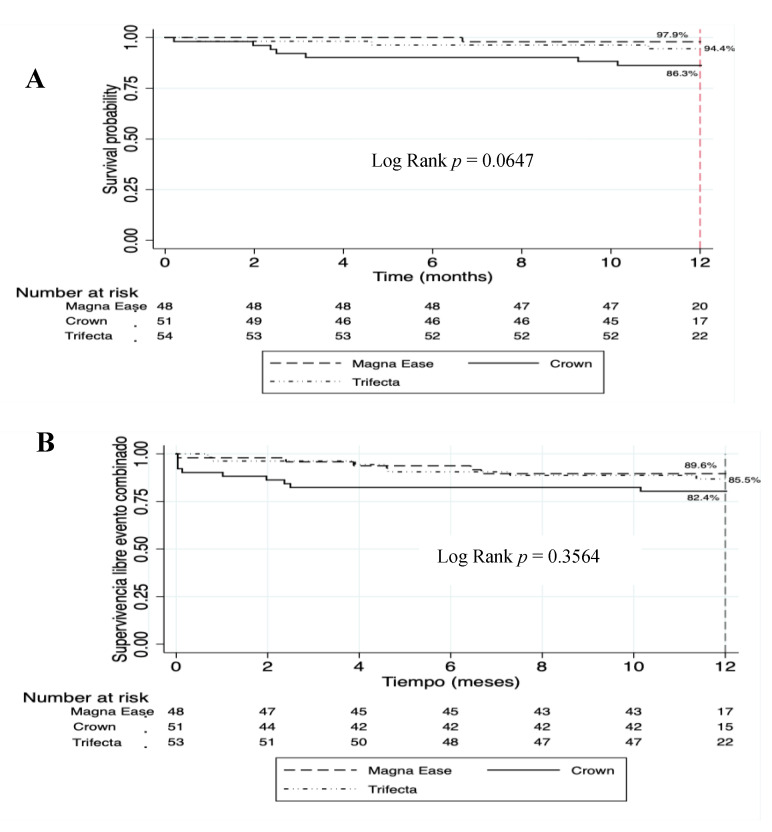
Kaplan–Meier survival graph (**A**) and Kaplan–Meier event-free survival graph by prosthesis type (**B**).

**Table 1 jcm-10-05340-t001:** Baseline characteristics and echocardiographic findings.

	Magna Ease^TM^ (*n* = 48)	Crown PRT^TM^ (*n* = 51)	Trifecta^TM^ (*n* = 55)	Global (*n* = 154)	*p*
BASELINE CLINICAL CHARACTERISTICS					
Male gender	34 (70.83%)	24 (47.06%)	34 (61.82%)	92 (59.74%)	0.051
Age (years)	76.5 (IQR 72.5–79)	76.5 (IQR 68.5–81.5)	77.5 (IQR 72.5 79.5)	76.5 (IQR 71.5–79.5)	0.916 Ψ
Body mass index	28.4 (26.1–31.1)	28.1 (24.9–30.8)	28.6 (27.0–31.4)	28.4 (25.5–31.2)	0.428 Ψ
Arterial hypertension	35 (72.92%)	40 (78.43%)	42 (76.36%)	117 (75.97%)	0.817 Ψ
Diabetes	20 (41.67%)	17 (33.33%)	17 (30.91%)	54 (35.06%)	0.497 Ψ
Smoking	1 (2.0%)	4 (7.84%)	5 (9.09%)	10 (6.49%)	0.670 Ψ
Dyslipemia	34 (70.83%)	34 (66.67%)	36 (65.45%)	104 (67.53%)	0.833
COPD	8 (16.67%)	5 (9.8%)	2 (3.64%)	15 (9.74%)	0.083 Ψ
Renal failure	7 (14.58%)	6 (11.76%)	6 (10.91%)	19 (12.34%)	0.866 Ψ
NYHA IV	14 (29.16%)	19 (37.35%)	14 (25.4%)	47 (30.52%)	0.932 Ψ
Elective surgery	41 (85.42%)	34 (66.67%)	34 (61.82%)	109 (70.78%)	0.020
Previous mayor CVA	2 (4.17%)	1 (1.96%)	2 (3.64%)	5 (3.25%)	0.866 Ψ
Previous MI	3 (6.25%)	4 (7.84%)	10 (18.18%)	17 (11.04%)	0.176 Ψ
EUROSCORE II (%)	2.0 (IQR 1.2–2.8)	2.3 (IQR 1.4–4.6)	2.4 (IQR 1.5–4.8)	2.3 (IQR 1.4–4)	0.125 Ψ
EUROSCORE I (%)	5.5 (IQR 4.3–7.5)	6.0 (IQR 4.1–8.4)	6.6 (IQR 5.1–8.4)	6.2 (IQR 4.4–8.5)	0.312 Ψ
PREOPERATIVE ECHOCARDIOGRAPHIC FINDINGS					
Severe aortic stenosis	44 (91.67%)	43 (84.31%)	48 (87.27%)	135 (87.66%)	0.225 Ψ
Severe aortic regurgitation	3 (6.25%)	7 (13.73%)	5 (9.09%)	15 (9.74%)	0.877 Ψ
PASP > 55 mmhg	1 (2.08%)	1 (1.96%)	0 (0.0%)	2 (1.3%)	0.604 Ψ
LVEF (%)	65 (IQR 60.3–72.2)	62.7 (IQR 59–70.7)	65 (IQR 57.7–65)	65 (IQR 59.2–71.8)	0.65 Ψ
Peak aortic gradient (mmHg)	74.2 (±20.4)	69.7 (±29.2)	71.4 (±22.5)	71.7 (±24.3)	0.658
Mean aortic gradient (mmHg)	42.4 (±12.6)	41.5 (±18.2)	42.5 (±14.9)	42.1(±15.34)	0.931
Peak aortic velocity (cm/s)	429.4 (IQR 395.3–458.9)	424 (IQR 363.8–480.4)	429 (IQR 388.6–478.1)	428 (IQR 388.6–472.4)	0.829 Ψ
Aortic valve area (cm^2^)	0.7 (IQR 0.6–0.9)	0.7 (IQR 0.6 – 0.9)	0.71 (IQR 0.6–0.9)	0.7 (IQR 0.6–0.9)	0.466 Ψ
Left ventricular mass (g)	220.8 (IQR 167.9–277)	232.8 (IQR 185.6–294)	236 (IQR 187.8–298.6)	226.6 (IQR 179.3–293.5)	0.377 Ψ
LVTSV (mL)	38.8 (IQR 25.4–48.7)	36.6 (IQR 26.6–50.1)	35.9 (IQR 22.6–49.3)	36.1 (IQR 24–49)	0.892 Ψ
LVTDV (mL)	93.4 (IQR 64.1–124.7)	97.3 (IQR 73.3–128.7)	96.4 (IQR 74.5–117.8)	96.8 (IQR 70.8 124.7)	0.866 Ψ
TAPSE (cm)	2.2 (IQR 2–2.4)	2.2 (IQR 1.9–2.3)	2.1 (IQR 2–2.4)	2.2 (IQR 2–2.4)	0.545 Ψ
PROCEDURAL CHARACTERISTICS					
Mini-sternotomy	11 (22.92%)	8 (15.69%)	4 (7.27%)	23 (14.94%)	0.083
CBP time (min)	78.5 (IQR 61.5–92.5)	73 (IQR 56–94)	76 (IQR 65–98)	75 (IQR 61–94)	0.624 Ψ
Aortic cross-clamp time (min)	61 (IQR 51–78)	57 (IQR 47–67)	61 (IQR 51–81)	60 (IQR 50–77)	0.352 Ψ
Hegar sizer (mm)					0.892
≤22	20 (41.67%)	28 (54.90%)	28 (50.90%)	76 (49.39%)	
>22	28 (58.33%)	23 (45.14%)	27 (49.09%)	78 (50.64%)	
Prostheses sizer (mm)					0.894
≤21	24 (50.00%)	28 (54.90%)	29 (52.72%)	81 (52.60%)	
>21	24 (50.00%)	23 (45.09%)	26 (47.27%)	73 (47.40%)	
CABG	9 (18.75%)	12 (23.53%)	15 (27.27%)	36 (23.38%)	0.604 Ψ

Continuous variables were summarized with the mean ± standard deviation or median and interquartile. Categorical variables were summarized with absolute and relative frequency (%). Ψ: nonparametric tests on variables that do not follow a normal distribution. CABG: coronary artery bypass grafting, CBP: cardiopulmonary bypass, CVA: cardiovascular accident, COPD: chronic obstructive pulmonary disease, LTSV: left ventricular telesystolic volume, LVEF: left ventricular ejection fraction, LVTDV: left ventricular telediastolic volume, MI: myocardial infarction, PSAP: pulmonary artery systolic pressure, TAPSE: tricuspid annular plane systolic excursion.

**Table 2 jcm-10-05340-t002:** Echocardiographic findings 12 months after surgery.

	Magna Ease^TM^ (*n =* 47)	Crown PRT^TM^ (*n* = 44)	Trifecta^TM^ (*n* = 51)	Global (*n* = 142)	*p*	*p*-Value		
						Crown vs. Magna Ease	Trifecta vs. Magna Ease	Trifecta vs. Crown
Peak aortic gradient (mmHg)	17.5 (IQR 11.3–26)	21.4 (IQR 14.5–26.7)	13 (IQR 9.6–17.8)	16.9 (IQR 11.1–25)	0.003	0.90	0.51	0.03
Mean aortic gradient (mmHg)	7.8 (IQR 6.1–12)	10.4 (IQR 7–13.3)	6.6 (IQR 4.8–8.6)	7.75 (IQR 5.3–11.9)	0.003	0.77	0.51	0.03
Peak aortic velocity (cm/s)	227.1 (IQR 202.0 –268.8)	237.8 (IQR 195.9 –261.9)	209.7 (IQR 176.5 –241.4)	222.6 (IQR 194.1–259.2)	0.025	1.00	0.33	0.38
Effective orifice area (cm^2^)	1.4 (IQR 1.3–1.7)	1.4 (IQR 1.2–1.7)	1.65 (IQR 1.4–2)	1.55 (IQR 1.2–1.8)	0.242	-	-	-
LVEF %	62.9 (IQR 60–67.9)	61 (IQR 58.6–65.7)	60.7 (IQR 56.1–66.9)	61.5 (IQR 57.2–67.5)	0.738	-	-	-
LV mass (g)	168.7 (IQR 134.7–214)	184.7 (IQR 147.8 –229.9)	191 (IQR 143.9–230.2)	182.3 (IQR 144.5–230.1)	0.533	-	-	-
LVTSV (mL)	26.8 (IQR 20.1–38.5)	34 (IQR 24–39.4)	30.6 (IQR 20–43)	31.5 (IQR 20.4–48)	0.839	-	-	-
LVTDV (mL)	80.9 (IQR 61.7–121.1)	79.6 (IQR 59.5–97.8)	79.4 (IQR 60.8–102.5)	80.1 (IQR 60.8–105.8)	0.599	-	-	-
TAPSE (cm)	1.8 (IQR 1.7–1.9)	1.7 (IQR 1.6–2.0)	1.8 (IQR 1.6–2.0)	1.7 (IQR 1.6–1.9)	0.467	-	-	-

Continuous variables were summarized with the mean ± standard deviation or median and interquartile. Categorical variables were summarized with absolute and relative frequency (%). Nonparametric tests were performed. LTSV: left ventricular telesystolic volume, LVEF: left ventricular ejection fraction, LVTDV: left ventricular telediastolic volume, MI: myocardial infarction, TAPSE: tricuspid annular plane systolic excursion.

**Table 3 jcm-10-05340-t003:** Echocardiographic findings stratified by prosthetic size at 12 months after surgery.

Size		Magna Ease^TM^	Crown PRT^TM^	Trifecta^TM^	*p*	*p*-Value		
						Crown vs. Magna Ease	Trifecta vs. Magna Ease	Trifecta vs. Crown
Nº ≤ 21		*n* = 23	*n* = 25	*n* = 29				
	Peak aortic gradient (mmHg)	17.8 (IQR 13.6–25.8)	26 (IQR 19.4–34.3)	12.3 (IQR 9.3–19.9)	0.001 Ψ	0.216	0.551	0.004
	Mean aortic gradient (mmHg)	8 (IQR 6.9–12.0)	12 (IQR 9.4–18.0)	6.7 (IQR 4.3–10.5)	0.004 Ψ	0.204	1.000	0.014
	Peak aortic velocity (cm/s)	236.8 (IQR 209–267.8)	251.4 (IQR 228.5–271.5)	209.7 (IQR 176.5–241.9)	0.033 Ψ	1.000	0.576	0.079
	Effective orifice area (cm^2^)	1.4 (IQR 1.2–1.6)	1.2 (IQR 1.2–1.3)	1.55 (IQR 1.3–1.8)	0.287 Ψ	-	-	-
Nº > 21		*n* = 24	*n* = 19	*n* = 22				
	Peak aortic gradient (mmHg)	17.5 (IQR 11–27)	16 (IQR 9.2–21.9)	13.3 (IQR 10.2–17.8)	0.455 Ψ	-	-	-
	Mean aortic gradient (mmHg)	7.5 (IQR 5.4–13.0)	8.6 (IQR 5.1–11.4)	6.6 (IQR 4.9–7.3)	0.254 Ψ	-	-	-
	Peak aortic velocity (cm/s)	223.4 (IQR 194.4–270.7)	217.3 (IQR 182.5–252.5)	209.7 (IQR 175.1–229.9)	0.274 Ψ	-	-	-
	Effective orifice area (cm^2^)	1.7 (IQR 1.5–2.4)	1.65 (IQR 1.5–1.8)	1.8 (IQR 1.6–2.1)	0.527 Ψ	-	-	-

Ψ: nonparametric tests on variables that do not follow a normal distribution.

**Table 4 jcm-10-05340-t004:** Echocardiographic finding stratified by aortic annulus size at 12 months after surgery.

Size		Magna Ease^TM^	Crown PRT^TM^	Trifecta^TM^	*p*	*p*-Value		
						Crown vs. Magna Ease	Trifecta vs. Magna Ease	Trifecta vs. Crown
Nº ≤ 22		*n* = 19	*n* = 24	*n* = 26				
	Peak aortic gradient (mmHg)	17.8 (IQR 13.6–26.5)	26 (IQR 19.4–28.7)	13.6 (IQR 10.2–17.0)	0.001	0.467	0.479	0.006
	Mean aortic gradient (mmHg)	8.95 (IQR 7.0–12.3)	12.6 (IQR 9.4–18)	6.6 (IQR 4.9–9.1)	0.003	0.435	0.840	0.018
	Peak aortic velocity (cm/s)	236.8 (IQR 209–269)	256.8 (IQR 230.2–271.5)	207.7 (IQR 184.2–241.9)	0.014	0.920	0.592	0.048
	Effective orifice area (cm^2^)	1.4 (IQR 1.2–1.7)	1.2 (IQR 1.2–1.3)	1.5 (IQR 1.3–1.6)	0.304	-	-	-
Nº > 22		*n* = 28	*n* = 20	*n* = 25				
	Peak aortic gradient (mmHg)	17.5 (IQR 11–25.5)	16 (IQR 9.4–21.5)	12.7 (IQR 9.3–19.8)	0.376	-	-	-
	Mean aortic gradient (mmHg)	7.5 (IQR 5.4–11.9)	8.5 (IQR 5.1–11.1)	6.6 (IQR 4.5–8.2)	0.313	-	-	-
	Peak aortic velocity (cm/s)	224.8 (IQR 194.4–266.7)	210.8 (IQR 180–247.6)	210 (IQR 176.5–230.4)	0.197	-	-	-
	Effective orifice area (cm^2^)	1.6 (IQR 1.3–2.4)	1.7 (IQR 1.5–1.8)	1.8 (IQR 1.6–2.1)	0.650	-	-	-

Continuous variables were summarized with the mean ± standard deviation or median and interquartile. Categorical variables were summarized with absolute and relative frequency (%).

**Table 5 jcm-10-05340-t005:** Postoperative events at 12 months.

	Magna Ease^TM^ (*n* = 48)	Crown PRT^TM^ (*n* = 51)	Trifecta^TM^ (*n* = 55)	Global (*n* = 154)	*p*
Myocardial infarction	0 (0.00%)	4 (7.84%)	3 (5.45%)	7 (4.55%)	0.218
Cerebrovascular events	2 (4.17%)	3 (5.88%)	2 (3.64%)	7 (4.55%)	0.880
Thromboembolic complications	0 (0.00%)	0 (0.00%)	0 (0.00%)	0 (0.00%)	-
Prosthetic valve deterioration	0 (0.00%)	0 (0.00%)	1 (1.82%)	1 (0.65%)	-
Endocarditis	2 (4.17%)	0 (0.00%)	2 (3.64%)	4 (2.60%)	0.910
Reintervention	0 (0.00%)	0 (0.00%)	1 (1.82%)	1 (0.65%)	-
Event-free survival	43 (89.6%)	42 (82.4%)	47 (85.5%)	132 (85.7%)	0.356

Continuous variables were summarized with the mean ± standard deviation or median and interquartile. Categorical variables were summarized with absolute and relative frequency (%).

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
