# Peer review of "One-Year Hemodynamic Performance of Three Cardiac Aortic Bioprostheses: A Randomized Comparative Clinical Trial"

_jcm, 2021, doi:10.3390/jcm10225340_

Round 1
Reviewer 1 Report
The authors aimed to compare 1 year the hemodynamic in-vivo performance of 3 biological aortic prostheses (Carpentier Perimount Magna EaseTM, Crown PRTTM and TrifectaTM). Methods: The sample used in this study comes from the “BEST-VALVE” clinical trial, which is a phase IV single blinded randomized clinical trial with 3 above mentioned prostheses.
154 patients (aged over 18 years, median age 76.5 yrs, 92 males) were included (from June 2014-June 2017), with aortic stenosis or insufficiency, needing valve replacement with random assignment 1:1:1 - Carpentier Perimount Magna EaseTM (n=48, 31.2%), Crown PRTTM (n=51,32.1%) and TrifectaTM (n=55, 35.7%). The primary endpoint was to compare hemodynamic performance (aortic mean and peak gradients and effective orifice area) and a secondary endpoint was to compare the 3 protheses including survival at 12 months follow-up. The continuity equation (Gorlin’s formula) was used to estimated the effective orifice area.
Results: One year after the surgery, the mean aortic gradient and the peak aortic velocity was 17.5 (IQR 11.3 – 26) and 227.1cm/s (IQR 202.0 – 268.8) for Carpentier Perimount Magna EaseTM, 21.4 (IQR 14.5 – 26.7) and 237.8 cm/s (IQR 195.9 – 261.9) for Crown PRTTM, and 13 (IQR 9.6 – 17.8) and 209.7 cm/s (IQR 176.5 – 241.4) for TrifectaTM.
Pairwise comparisons demonstrated improved mean gradients and maximum velocity of TrifectaTM as compared to Crown PRTTM. Among patients with nominal prosthesis sizes ≤21, the mean and peak aortic gradient was higher for Crown PRTTM comparing with TrifectaTM, and also in patients with an aortic annulus measured with a metric Hegar dilators less than or equal to 22mm.
No statistically significant differences were found between the 3 protheses as for the incidence of postoperative events at 12 months after procedure.
In conclusion, the authors state that one year after surgery, the 3 prostheses presented a different hemodynamic performance, being TrifectaTM superior to Crown PRTTM.
Opinion of the reviewer:
The paper is well written, well documented with tables and figures; the discussion is appropriate.
The limitation of the study- as the authors state, is that it is a single center study with a limited sample size, the results could not be comparable with those obtained in other centers.
However, I think the paper is valuable, useful for the operators in the field
Author Response
We want to thank you for your comments about our study.
Reviewer 2 Report
Great study, a well done RCT
- - state if it is a trial funded by the manufacturers of one of the valves in the study - Excellent echocardiographic data collection - Put the pictures of the 3 valves - There are magnetic resonance studies that evaluate the flowmetry of valves (they are small studies but they deserve to be mentioned, in the future the CMR could become the gold standard)? - Report at least one anthropometric value in the patient variable - KM curves well done, perhaps they should be enlarged by adapting the value on the Y axis - Why are there different gradients with similar valve area values? - Prognostically is the gradient or the EROA more important? - CT scans are increasingly important in echocardiographic calculations and also in surgical planning (as in the TAVI world) - cite this article on the use of imaging in the TAVI world in valve selection (DOI: 10.23736 / S2724-5683.21.05573-0)
Author Response
We want to thank you for your comments.
- State if it is a trial funded by the manufacturers of one of the valves in the study: The trial was not funded by any manufacturer.
- Excellent echocardiographic data collection: We want to thank you for your comments.
- Put the pictures of the 3 valves: the pictures of the three aortic bioprostheses have been included.
- There are magnetic resonance studies that evaluate the flowmetry of valves (they are small studies but they deserve to be mentioned, in the future the CMR could become the gold standard)?: We have included some articles related to magnetic resonance imaging and cardiac valvular evaluation.
- Report at least one anthropometric value in the patient variable: Body mass index has been included.
- KM curves well done, perhaps they should be enlarged by adapting the value on the Y axis: we will consider it.
- Why are there different gradients with similar valve area values?.
There is not a direct relationship between gradients and valves areas, because these parameters can be influenced by other variables. Transvalvular gradients are highly flow dependent. On the other hand, aortic valve area is a parameter less dependent on flow and more dependent on the measurement of the left ventricular outflow tract, the main source of error. There are different recent studies that show that patients with severe aortic stenosis with aortic valve area less than 1 cm2 with preserved left ventricular ejection fraction have a lack of correlation between flow and gradients.
- Minners, M. Allgeier, C. Gohlke-Baerwolf, R.R. Kienzle, F.J. Neumann, N. Jander. Inconsistencies of echocardiographic criteria for the grading of aortic valve stenosis. Eur Heart J, 29 (2008), pp. 1043-1048.
M.A. Clavel, J.G. Dumesnil, R. Capoulade, P. Mathieu, M. Sèneèchal, P. Pibarot. Outcome of patients with aortic stenosis, small valve area, and low-flow, low-gradient despite preserved left ventricular ejection fraction. J Am Coll Cardiol, 60 (2012), pp. 1259-1267
- Hachicha, J.G. Dumesnil, P. Bogaty, P. Pibarot. Paradoxical low flow, low gradient severe aortic stenosis despite preserved ejection fraction is associated with higher after load and reduced survival. Circulation, 115 (2007), pp. 2856-2864
- Prognostically is the gradient or the EROA more important?.
The objective of our study is not to quatify whether EROA has a greater relationship with clinical events than valve gradients, and also our study could have not sufficient statistical power.
With the results of our study we can affirm that at 12 months after surgery the different types of prostheses differ, presenting statistically significant differences with respect to the mean and peak gradients between Crown PRT and Trifecta.
- CT scans are increasingly important in echocardiographic calculations and also in surgical planning (as in the TAVI world) cite this article on the use of imaging in the TAVI world in valve selection (DOI: 10.23736 / S2724-5683.21.05573-0). This reference has been included in the article.